# Experiences and well-being of healthcare professionals working in the field of ultrasound in obstetrics and gynaecology as the SARS-CoV-2 pandemic were evolving: a cross-sectional survey study

Tom Bourne [1,2] Christopher Kyriacou [1] Harsha Shah [1] Jolien Ceusters,[3] Jessica Preisler,[4,5] Ulrike Metzger,[6] Chiara Landolfo [7] Christoph Lees [8] Dirk Timmerman [2,9]

For numbered affiliations see end of article.

**Correspondence to**
Professor Tom Bourne;
t.bourne@imperial.ac.uk

## ABSTRACT

**Objective** Assess experience of healthcare professionals (HCPs) working with ultrasound in obstetrics and gynaecology during the evolving SARS-CoV-2 pandemic, given the new and unprecedented challenges involving viral exposure, personal protective equipment (PPE) and well-being.

**Design** Prospective cross-sectional survey study.

**Setting** Online international survey. Single-best, open box and Hospital Anxiety and Depression Scale (HADS) questions.

**Participants** The survey was sent to 35 509 HCPs in 124 countries and was open from 7 to 21 May 2020. 2237/3237 (69.1%) HCPs from 115 countries who consented to participate completed the survey. 1058 (47.3%) completed the HADS.

**Primary outcome measures** Overall prevalence of SARS-CoV-2, depression and anxiety among HCPs in relation to country and PPE availability.

**Analyses** Univariate analyses were used to investigate associations without generating erroneous causal conclusions.

**Results** Confirmed/suspected SARS-CoV-2 prevalence was 13.0%. PPE provision concerns were raised by 74.1% of participants; highest among trainees/resident physicians (83.9%) and among HCPs in Spain (89.7%). Most participants worked in self-perceived high-risk areas with SARS-CoV-2 (67.5%–87.0%), with proportionately more trainees interacting with suspected/confirmed infected patients (57.1% vs 24.2%–40.6%) and sonographers seeing more patients who did not wear a mask (33.3% vs 13.9%–7.9%). The most frequent PPE combination used was gloves and a surgical mask (22.3%). UK and US respondents reported spending less time self-isolating (8.8 days) and lower satisfaction with their national pandemic response (37.0%–43.0%). 19.8% and 8.8% of respondents met the criteria for moderate to severe anxiety and depression, respectively.

**Conclusions** Reported prevalence of SARS-CoV-2 in HCPs is consistent with literature findings. Most respondents

## Strengths and limitations of this study

► Strengths of our study include the sample size, the assessment of healthcare professional (HCP) responses internationally and the ability to compare responses between different HCPs.

► Four languages enabled greater participant inclusion, and responses between International Society of Ultrasound in Obstetrics and Gynecology (ISUOG) and non-ISUOG members were compared prior to grouping for analysis.

► Key weaknesses include possible selection bias due to the method of sample selection and the low population response rate.

► It is important not to draw conclusions about causation as this is a descriptive study performed as the pandemic was evolving.

► At the time of survey participation, the majority of HCPs did not have the same access to a swab or antibody test that is now available.

used gloves and a surgical mask, with a greater SARS-CoV-2 prevalence compared with those using 'full' PPE. HCPs with the least agency (trainees and sonographers) were not only more likely to see high-risk patients but also less likely to be protected. A fifth of respondents reported moderate to severe anxiety.

## INTRODUCTION
### Background
The SARS-CoV-2 pandemic has caused unprecedented challenges to all healthcare systems.[1] These include patient protection, managing the burden of disease on service capacity and mitigating adverse effects on healthcare professionals (HCPs: physicians, sonographers, nurses and allied HCPs).[2]

Guidance has internationally focused on clinician redeployment to front-line settings, prioritisation of services, provision and rationalisation of personal protective equipment (PPE).[3 4] During the pandemic, access to PPE for HCPs has remained a key concern. Despite HCPs being prioritised in most countries, PPE shortages have been described universally[1] and have been a particular concern in the early months of the pandemic. Lack of sufficient and adequate PPE is important in obstetrics and gynaecology, where HCPs work in a variety of settings, including ultrasound scanning, may not ordinarily be considered high-risk but involves close patient proximity for extended periods.[5] These areas are easily overlooked but may pose a high risk of SARS-CoV-2 infection.[6]

A further concern has been the availability of SARS-CoV-2 testing among HCPs. Initially, provision for this was variable, with, for example, relatively easy access in Hong Kong compared with virtually none in the UK. This is important as, in the absence of testing, HCP risk becoming COVID-19 vectors within hospitals and the wider community. Lack of testing may also lead to unnecessary isolation from work. There are limited data regarding HCP testing availability and time HCPs have taken off work due to suspected or confirmed SARS-CoV-2.[7]

Less attention has been focused on the implications of infectious diseases on HCP psychological health, which previous pandemics have shown to be significant.[8 9] HCPs are confronted with ongoing resource, shift pattern, PPE and testing uncertainty,[1 10] as well as COVID-19 exposure risk and the implications of this on their families.[11] Protecting the psychological health of the medical workforce is critical, particularly as anxiety, depression and burnout are recognised complications for HCPs working in high-stress environments.[12]

## Objectives

We aimed to assess the scale, experience and psychological well-being of clinicians who have experienced the evolving SARS-CoV-2 pandemic working with ultrasound in obstetrics and gynaecology internationally between 7 and 21 May 2020.

## METHODS
### Study design

We used a cross-sectional survey design with two main sections assessing physical aspects, including underlying medical conditions, comorbidities and personal SARS-CoV-2 experience, and psychological well-being (online supplement 1: The International Society of Ultrasound in Obstetrics and Gynecology (ISUOG) SARS-CoV-2). Participants were initially asked to provide informed consent prior to continuing to the online survey. If they did not give consent by answering 'yes' to the first question, they were not directed to the survey. Single-best answer and open box questions were proposed by a senior gynaecologist. These were then piloted, vetted and modified by the authors before being translated from English to French,

Spanish and Italian in order to increase survey accessibility. We estimated that the time taken to complete the questionnaire was 10 min.

### Setting

The online platform SurveyMonkey (2020) was selected to upload and disseminate the questionnaire, which was encrypted, and multiple responses were disabled. The survey design ensured the data collected was completely anonymous for all respondents.

### Participants

All HCPs on the International Society of Ultrasound in Obstetrics and Gynaecology (ISUOG) mailing list were invited to participate in this study between 7 and 21 May 2020. There were 1 237 953 confirmed cases of SARS-CoV-2 infection and 63 313 deaths in the time the survey was open.[13] Various countries had instigated local and national lockdowns at this time, including the UK. Members were sent an email containing information describing the study and individual links to the questionnaire in English, Spanish, Italian and French. We made it clear to the participants that their participation was voluntary and that responses would be both anonymous and untraceable.

During the survey period, two reminders were sent out. As this was a survey study of a relatively new condition at the time, a power calculation was not performed. Exclusion criteria included retired doctors as this was deemed to not be reflective of the working environment during the SARS-CoV-2 pandemic. A copy of the questionnaire in English can be viewed in online supplement 1: The International Society of Ultrasound in Obstetrics and Gynecology (ISUOG) SARS-CoV-2.

### Patient and public involvement statement

Given the HCP participant population and the acute nature of the SARS-CoV-2 pandemic, patient and public involvement was not performed.

### Variables and data sources
#### Physical aspects

This section was divided into subsections: Introduction; Demographics; Underlying medical conditions; Medications; Smoking and BMI; Role, shift pattern, PPE; SARS-CoV-2 diagnosis; Personal SARS-CoV-2 pathway; Household; Colleagues; and Support (online supplement 1: The International Society of Ultrasound in Obstetrics and Gynecology (ISUOG) SARS-CoV-2: questions 2–40).

Participants were asked details about their job and organisational attributes, as well as on a variety of personal comorbidities relevant to SARS-CoV-2 infection.[14] They were then questioned on their personal experience of SARS-CoV-2 at home and in the workplace.

### Psychological well-being - Hospital Anxiety and Depression Scale (HADS)

Participants were given the option to complete the validated HADS, which is a 14-item (seven questions related

to anxiety and depression each) questionnaire[15] (online supplement 1: The International Society of Ultrasound in Obstetrics and Gynecology (ISUOG) SARS-CoV-2: questions 41–56). Each subscale measures symptom severity (the score ranges between 0 and 21, a score greater or equal to 11 indicates moderate-to-severe symptoms). If respondents chose not to complete the HADS, the survey ended.

## Outcome measures

The primary outcome was the overall prevalence of SARS-CoV-2, depression and anxiety among HCPs in relation to country and PPE availability.

## Statistical analyses

Univariate analyses were used to investigate the association between country of origin, type of HCP and PPE without generating erroneous causal conclusions. Results were presented as percentages (%), mean, 95% CIs, medians and ranges, depending on the variable. Missing data, that is, questions that were skipped by HCPs as they completed their questionnaire, were not included in the analyses.

HCPs and areas of work were divided into groups to allow reasonable comparable analyses to be performed. For HCPs, there were three main categories: consultants/attending physicians (gynaecologists, obstetricians, obstetricians and gynaecologists or radiologists/sonologists aged 35 years and older); trainees/residents (gynaecologists, obstetricians, obstetricians and gynaecologists or radiologists/sonologists aged 34 years and younger, doctors in research and doctors in training); and sonographers (sonographers and allied HCPs (midwives and nurses)) (online supplement 3: Breakdown of healthcare professional (HCP) roles for analysis).

Areas of work were defined as high-risk (gynaecology (early pregnancy unit, acute gynaecology and ultrasound), obstetrics (birth centre/birthing centre, labour ward, maternity triage/hospital obstetric triage and ultrasound) and radiology (sonology–ultrasound)); moderate risk (obstetrics (antenatal clinic, antenatal ward, community clinics and postnatal ward) and radiology (non-ultrasound)); or low-risk (working from home, gynaecology (benign and oncology) and obstetrics (maternity helplines))) (online supplement 4: Breakdown of risk by area of work for analysis). This categorisation approximated risk based on HCP proximity and time spent with patients, allowing meaningful data analysis.

Anxiety and depression scores were divided by caseness in order to differentiate the proportion of the study population that had no anxiety or depression (HADS score 0–7), those experiencing mild anxiety or depression (HADS score 8–10), and those suffering from moderate-to-severe anxiety or depression (HADS 11–21).

The statistical analyses were performed using R V.3.5.1.

# RESULTS
## Summary
### Response

The survey was sent to 35 509 HCPs in 124 countries. A total of 3287 (9.3%) HCPs clicked the survey link and 3237 (98.5%) of them consented to complete it. A total of 1960 (60.5%) responded in English, 881 (27.2%) in Spanish, 267 (8.2%) in Italian and 129 (4.0%) in French. The response rate for ISUOG members was 23.1% (2441/10589), while that for non-ISUOG members was 3.2% (796/24920) (online supplement 5: Flowchart summarizing survey participants, differentiating International Society of Ultrasound in Obstetrics and Gynecology (ISUOG) members and non-ISUOG members (n = 3287)). Of 3237 HCPs from 115 countries who started the survey, 2237 (69.1%) completed it in full. Mean age of respondents was 47.2 years, with a range from 18 to 82 years. The majority (83%) were aged 31–60 years old. A total of 1474 (66%) of respondents were female and 755 (34%) were male, with 4 intersex or undisclosed. Of 2237 participants, 1058 (47.3%) completed the HADS, and the remainder (1179/2237, 52.7%) did not. Survey participation and demographics are summarised in table 1.

Direct comparison of data between the characteristics of ISUOG members and non-ISUOG members confirmed findings were similar (online supplement 6: Demographics of survey participation, differentiated based on International Society of Ultrasound in Obstetrics and Gynecology (ISUOG) membership (n = 3237)). It was therefore deemed appropriate to combine these populations for the analysis.

### COVID-19 prevalence

Confirmed or suspected SARS-CoV-2 prevalence was 13.0% (290/2237). Among this group, 11.4% had a history of heart/lung disease; 13.4% were taking antihypertensives; and 4.5% were smokers (online supplement 7: Comparison of comorbidity in those with (n = 290) and without suspected or confirmed SARS-CoV-2 (n = 290) (Total n = 2237)).

### PPE use and provision

Of all HCPs, 12.5% reported not using PPE, while 87.5% reported variable use. Of the total HCPs, 74.1% had concerns about provision or shortages of one or more items of PPE, in particular N95/FFP3 masks (57.2%). PPE was reported to be recycled or reused by 61.3%.

### Anxiety and depression

Of the total participants, 21.6% met the criteria for mild anxiety, and 19.8% met the criteria for moderate-to-severe anxiety; 19.4% met the criteria for mild depression and 8.8% for moderate-to-severe depression. Of those with suspected/confirmed SARS-CoV-2, 27.2% and 12.3% reported moderate-to-severe anxiety and depression, compared with 18.4% and 8.1% of those without (online supplement 8: Comparison of Hospital Anxiety and Depression scale (HADS) scores in those with (n =

**Table 1** Demographics of total survey participation (N=3237)

| | Total (N=3237) |
|---|---|
| Total ISUOG mailing list (n) | 35 509 |
| Response of total ISUOG mailing list, N (%) | 3237 (9) |
| Countries (n) | 124 |
| Respondents in English (n) | 1960 |
| Respondents in Spanish (n) | 881 |
| Respondents in Italian (n) | 267 |
| Respondents in French (n) | 129 |
| ISUOG members, n (%) | 2441 (23) |
| Completed survey, n (%) | 2237 (69) |
| ISUOG members, n (%) | 1795 (80) |
| Countries (n) | 115 |
| Age (years), mean (range) | 47.2 (18–82) |
| ≤20, n (%) | 1 (0) |
| 21–30, n (%) | 117 (5) |
| 31–40, n (%) | 558 (25) |
| 41–50, n (%) | 686 (31) |
| 51–60, n (%) | 596 (27) |
| 61–70, n (%) | 251 (11) |
| >70, n (%) | 24 (1) |
| Gender, n (%) | |
| Female | 1474 (66) |
| Male | 755 (34) |
| Intersex | 1 (0) |
| Prefer not to say | 3 (0) |
| Prevalence suspected/confirmed SARS-CoV-2 based on symptoms±PCR testing, n (%) | 290 (13) |
| Is PPE used by the respondents? n (%) | |
| Not yet | 279 (12) |
| Yes—for aerosol-generating procedures only | 133 (6) |
| Yes—for all patients | 1392 (62) |
| Yes—for suspected/positive patients only | 429 (19) |
| What PPE is used by the respondents? n (%) | |
| None | 76 (3) |
| Gloves only | 49 (2) |
| Gloves+surgical mask | 497 (22) |
| Gloves+surgical mask+visor | 152 (7) |
| Gloves+FFP3 mask | 95 (4) |
| Gloves+FFP3 mask+visor | 51 (2) |
| Gloves+FFP3 mask+visor+gown+surgical hat | 90 (4) |
| Patient wearing mask | 1841 (82) |
| Concerns regarding a lack of PPE, n (%) | |
| No | 579 (26) |
| Gloves | 278 (12) |
| Gown | 629 (28) |
| N95/FFP3 mask | 1279 (57) |
| Surgical hat | 233 (10) |
| Surgical mask | 820 (37) |
| Visor | 582 (26) |

Continued

**Table 1** Continued

| | Total (N=3237) |
|---|---|
| Surgical mask (patient) | 609 (27) |
| Type of PPE that has been reused or recycled, n (%) | |
| No | 865 (39) |
| Gloves | 34 (2) |
| Gown | 287 (13) |
| N95/FFP3 mask | 886 (40) |
| Surgical hat | 92 (4) |
| Surgical mask | 565 (25) |
| Visor | 621 (28) |
| Anxiety, mean (95% CI)* | 6.9 (6.7 to 7.2) |
| 0–7 (none), n (%) | 620 (59) |
| 8–10 (mild), n (%) | 229 (22) |
| 11–21 (moderate to severe), n (%) | 209 (20) |
| Depression, mean (95% CI)* | 5.3 (5.1 to 5.5) |
| 0–7 (none), n (%) | 760 (72) |
| 8–10 (mild), n (%) | 205 (19) |
| 11–21 (moderate to severe), n (%) | 93 (9) |

*Numbers of Hospital and Anxiety Depression Scale (HADS) respondents lower than the main survey (n=1058).
ISUOG, International Society of Ultrasound in Obstetrics and Gynecology; PCR, Polymerase Chain Reaction; PPE, personal protective equipment.

162) and without suspected or confirmed SARS-CoV-2 (n = 896) (Total n = 1058)). A greater proportion of female participants scored for moderate-to-severe anxiety (156/754, 20.7%) and depression (79/754, 10.5%) compared with male participants (41/303, 13.5% and 14/303, 4.6%) (online supplement 9: Hospital Anxiety and Depression scale (HADS) breakdown by gender (n = 1058)).

Four out of 2237 participants were excluded from the univariate analyses, having completed the survey following retirement.

### Univariate analysis by country (N=2233)
A breakdown of findings by countries that had most respondents is presented in table 2.

### COVID-19 prevalence
The highest prevalence of SARS-CoV-2 infection based on symptoms and/or a positive PCR swab was reported in the UK (31.1%), followed by Italy (14.9%), Spain (14.7%) and the USA (13.7%). Respondents from India and the Philippines reported a prevalence of 2.6% and 8.8%, respectively. Only 0.9% and 1.0% of Indian and UK participants, respectively, had an antibody test, compared with 52.9% and 60.3% of participants in Italy and Spain, respectively. Of participants from the USA and Spain who had a SARS-CoV-2 antibody test, 25.0% and 13.4%, respectively, had a positive result.

Participants in the UK reported spending an average of 8.8 days self-isolating away from the workplace, the same as those in the USA. Mean self-isolation time was only

**Table 2** Breakdown of findings by country (N=2233)

| | Italy | Spain | India | Philippines | UK | USA | Other* |
|---|---|---|---|---|---|---|---|
| Respondents, n (%) | 174 (8) | 136 (6) | 115 (5) | 114 (5%) | 103 (5) | 102 (5) | 1489 (67) |
| Prevalence of suspected/confirmed SARS-CoV-2 based on symptoms±PCR testing, n (%) | 26 (15) | 20 (15) | 3 (3) | 10 (9) | 32 (31) | 14 (14) | 185 (12) |
| Had SARS-CoV-2 antibody test, n (%) | 92 (53) | 82 (60) | 1 (1) | 21 (18) | 1 (1) | 16 (16) | 155 (10) |
| Had positive SARS-CoV-2 antibody, n (%) | 6 (7) | 11 (13) | 0 (0) | 0 (0) | 0 (0) | 4 (25) | 16 (10) |
| **Is PPE used by the respondents? n (%)** | | | | | | | |
| Not yet | 17 (10) | 28 (21) | 31 (27) | 6 (5) | 7 (7) | 5 (5) | 185 (12) |
| Yes—for aerosol-generating procedures only | 8 (5) | 6 (4) | 9 (8) | 7 (6) | 6 (6) | 3 (3) | 94 (6) |
| Yes—for all patients | 107 (61) | 33 (24) | 65 (57) | 96 (84) | 76 (74) | 83 (81) | 932 (63) |
| Yes—for suspected/positive patients only | 42 (24) | 69 (51) | 10 (9) | 5 (4) | 14 (14) | 11 (11) | 278 (19) |
| **What PPE is used by the respondents? n (%)** | | | | | | | |
| None | 0 (0) | 1 (1) | 1 (1) | 1 (1) | 1 (1) | 0 (0) | 72 (5) |
| Gloves only | 0 (0) | 0 (0) | 0 (0) | 0 (0) | 2 (2) | 0 (0) | 47 (3) |
| Gloves+surgical mask | 79 (45) | 42 (31) | 7 (6) | 0 (0) | 41 (40) | 45 (44) | 283 (19) |
| Gloves+surgical mask+visor | 6 (3) | 11 (8) | 6 (5) | 1 (1) | 17 (17) | 12 (12) | 99 (7) |
| Gloves+FFP3 mask | 12 (7) | 10 (7) | 8 (7) | 0 (0) | 0 (0) | 6 (6) | 59 (4) |
| Gloves+FFP3 mask+visor | 1 (1) | 4 (3) | 6 (5) | 2 (2) | 0 (0) | 3 (3) | 35 (2) |
| Gloves+FFP3 mask+visor +gown+surgical hat | 3 (2) | 5 (4) | 13 (11) | 23 (20) | 0 (0) | 1 (1) | 45 (3) |
| Patient wearing mask | 173 (99) | 133 (98) | 109 (95) | 111 (97) | 33 (32) | 91 (89) | 1191 (80) |
| **Concerns regarding a lack of PPE, n (%)** | | | | | | | |
| No | 35 (20) | 14 (10) | 52 (45) | 20 (18) | 46 (45) | 24 (24) | 388 (26) |
| Gloves | 25 (14) | 19 (14) | 5 (4) | 12 (11) | 1 (1) | 9 (9) | 207 (14) |
| Gown | 53 (30) | 45 (33) | 23 (20) | 54 (47) | 25 (24) | 23 (23) | 406 (27) |
| N95/FFP3 mask | 118 (68) | 110 (81) | 51 (44) | 84 (74) | 33 (32) | 61 (60) | 822 (55) |
| Surgical hat | 15 (9) | 10 (7) | 10 (9) | 15 (13) | 4 (4) | 8 (8) | 171 (11) |
| Surgical mask | 84 (48) | 62 (46) | 10 (9) | 21 (18) | 24 (23) | 45 (44) | 574 (39) |
| Visor | 51 (29) | 51 (38) | 11 (10) | 19 (17) | 19 (18) | 24 (24) | 407 (27) |
| Surgical mask (patient) | 44 (25) | 47 (35) | 10 (9) | 32 (28) | 13 (13) | 31 (30) | 432 (29) |
| **Type of PPE that has been reused or recycled, n (%)** | | | | | | | |
| No | 55 (32) | 31 (23) | 40 (35) | 25 (22) | 62 (60) | 12 (12) | 640 (43) |
| Gloves | 2 (1) | 3 (2) | 5 (4) | 2 (2) | 0 (0) | 1 (1) | 21 (1) |
| Gown | 15 (9) | 17 (13) | 23 (20) | 70 (61) | 3 (3) | 4 (4) | 155 (10) |
| N95/FFP3 mask | 82 (47) | 82 (60) | 59 (51) | 66 (58) | 10 (10) | 68 (67) | 519 (35) |
| Surgical hat | 2 (1) | 7 (5) | 1 (1) | 16 (14) | 1 (1) | 4 (4) | 61 (4) |
| Surgical mask | 83 (48) | 60 (44) | 7 (6) | 13 (11) | 14 (14) | 65 (64) | 323 (22) |
| Visor | 23 (13) | 48 (35) | 31 (27) | 56 (49) | 28 (27) | 28 (27) | 407 (27) |
| Mean days of self-isolation (range) | 11.3 (0–30) | 11.6 (0–30) | 17.3 (0–30) | 16.9 (0–30) | 8.8 (0–30) | 8.8 (0–30) | 8.4 (0–30) |
| Mean days of household self-isolating (range) | 21.1 (14–30) | 10.5 (2–15) | 18 (10–30) | 15.3 (7–30) | 11.1 (2–14) | 15 (2–30) | 12.2 (1–30) |
| Respondents satisfied with local unit SARS-CoV-2 response, n (%) | 89 (51) | 97 (71) | 89 (77) | 89 (78) | 75 (73) | 74 (73) | 1018 (68) |
| Mean % level of respondent satisfaction with government SARS-CoV-2 response (95% CI) | 38 (34 to 41) | 23 (19 to 26) | 61 (56 to 65) | 48 (44 to 52) | 43 (38 to 48) | 37 (31 to 43) | 53 (52 to 55) |
| Anxiety, mean (95% CI)† | 6.3 (5.6 to 7.1) | 6.9 (6.1 to 7.7) | 7.4 (6.1 to 8.8) | 7.1 (5.6 to 8.6) | 7.2 (6.2 to 8.2) | 7.3 (6.1 to 8.4) | 6.9 (6.6 to 7.2) |
| 0–7 (none), n (%) | 74 (71) | 51 (61) | 22 (51) | 18 (50) | 36 (58) | 35 (59) | 384 (57) |
| 8–10 (mild), n (%) | 13 (12) | 21 (25) | 13 (30) | 10 (28) | 12 (19) | 8 (14) | 152 (23) |
| 11–21 (moderate to severe), n (%) | 17 (16) | 12 (14) | 8 (19) | 8 (22) | 14 (23) | 16 (27) | 134 (20) |
| Depression, mean (95% CI)† | 5.8 (5.2 t 6.5) | 5 (4.2 t 5.7) | 5.1 (4.1 to 6.1) | 5.3 (4.2 to 6.5) | 5.3 (4.3 to 6.2) | 4.8 (3.8 to 5.7) | 5.3 (5.0 to 5.6) |

**Table 2**   Continued

|  | Italy | Spain | India | Philippines | UK | USA | Other* |
|---|---|---|---|---|---|---|---|
| 0–7 (none), n (%) | 74 (71) | 67 (80) | 31 (72) | 27 (75) | 44 (71) | 46 (78) | 471 (70) |
| 8–10 (mild), n (%) | 20 (19) | 11 (13) | 10 (23) | 8 (22) | 12 (19) | 9 (15) | 135 (20) |
| 11–21 (moderate to severe), n (%) | 10 (10) | 6 (7) | 2 (5) | 1 (3) | 6 (10) | 4 (7) | 64 (10) |

*The 1489 within the 'other' cohort make up data from participants of the remaining 109 countries. We provide a breakdown of the six countries that had the most respondents.
†Numbers of Hospital and Anxiety Depression Scale (HADS) respondents lower than the main survey (n, Italy=104; Spain=84; India=43; Philippines=36; UK=62; USA=59; Other=670).
PCR, Polymerase Chain Reaction; PPE, personal protective equipment.

greater than 2 weeks in India and in the Philippines (17.3 and 16.9 days, respectively), where SARS-CoV-2 prevalence among HCPs in the study was lowest at the time of the study.

## PPE use and provision

The use of PPE when performing ultrasound was reported by 73.0%–95.1% of participants. The most common combination of PPE used while performing ultrasonography was gloves and a surgical mask, with the highest rates of use in Italy (45.4%), the UK (39.8%) and the USA (44.1%). No participants from the Philippines and only 6.1% from India reported using only gloves and a surgical mask. Visor use, in addition to gloves and a surgical mask, was highest in the UK (16.5%) and the USA (11.8%). Full PPE (gloves, FFP3/N95 mask, visor, gown and surgical hat) was used by 4% (90/2233) of the respondents. Full PPE use was highest in the Philippines (20.2%) and lowest in the UK (0%). Apart from the UK (32.0%), high rates of patients wearing a mask were noted in every other country analysed (80.0%–99.4%).

PPE provision or supply concerns were highest in Spain (89.7%), Italy (79.9%) and the Philippines (82.5%), and lowest in the UK (55.3%) and India (54.8%). Less PPE was reported as being recycled or reused in the UK (39.8%) compared with any other country analysed. In the USA, 88.2% of HCPs reported PPE recycling or reuse.

## Anxiety, depression and support

The HADS revealed that in the UK, 22.6%, and in the USA, 27.1% of participants met the criteria for moderate to severe anxiety, higher than any other country analysed, with Spanish participants having the lowest rate (14.3%). Italian, UK and 'other countries' participants had the highest rates of moderate to severe depression (9.6%–9.7%), with Philippines HCPs having the lowest rates (2.8%).

Levels of satisfaction with the support offered by local units during the pandemic ranged from 68% to 78% in all countries except Italy, where the level was markedly lower (51.1%). Indians had the highest satisfaction rate with their government response to the pandemic with a mean of 61%, compared with 43% in the UK, 37% in the USA, 38% in Italy and 23% in Spain.

## Univariate analysis by HCP (N=2233)

Table 3 focuses on HCPs—66.3% consultants/attending physicians, 22.3% sonographers and 11.4% trainee/resident physicians (who are the youngest cohort overall). Of physicians and sonographers, 62.4%–63.3% and 54.3%–55.6%, respectively, reported changes to their duties and patient contact throughout the pandemic.

## COVID-19 prevalence

Suspected or confirmed SARS-CoV-2 prevalence was similar across the groups (12.6% to 14.2%). 87.0% of trainees worked in high-risk areas, compared with 77.3% of consultants and 67.5% of sonographers. Trainees self-isolated for 1 day less on average (8.5 days) compared with consultants and sonographers (9.6 and 9.7 days, respectively).

## PPE use and provision

The PPE reported in each HCP group as most frequently used when performing ultrasonography was gloves and a surgical mask (21.0%–30.3%), with 11.6%–15.0% reporting no PPE use. Sonographers saw more patients who did not wear a mask (33.3%) compared with trainees (7.9%) and consultants (13.9%).

A larger proportion of trainees (57.1%) interacted with patients with suspected or confirmed SARS-CoV-2 compared with consultants (40.6%) and sonographers (24.2%) but were less likely to use PPE for all patients (48.8%), compared with consultants (63.1%) and sonographers (66.9%).

Of the total trainees, 83.9% reported PPE concerns (compared with 73.1% of consultants and 71.9% of sonographers), with 74.0% reusing or recycling PPE (compared with 59.9% consultants and 58.9% sonographers).

## Anxiety, depression and support

Rates of anxiety (39.4%–47.7%) and depression (26.9%–31.0%) were similar across the HCP groups, with 18.3%–25.2% and 7.5%–9.2% of participants meeting the criteria for moderate to severe anxiety and depression, respectively. Those working in high-risk areas appear more inclined to complete the HADS (online supplement 10: Hospital Anxiety and Depression scale (HADS) breakdown by risk of working area (n = 1058)). However, rates of mild to severe anxiety are overall similar, with more depression reported by those working in lower-risk areas.

**Table 3** Breakdown of findings by HCP (N=2233)

| | Consultants/attending | Sonographer | Trainees/residents |
|---|---|---|---|
| Respondents, n (%) | 1480 (66) | 499 (22) | 254 (11) |
| Changes in shift pattern as a result of the pandemic, n (%) | | | |
| Unable to work | 34 (2) | 21 (4) | 11 (4) |
| Change in duties due to SARS-CoV-2 pandemic, n (%) | | | |
| Increased work | 185 (13%) | 64 (13%) | 37 (15%) |
| Similar work | 570 (39%) | 228 (46%) | 82 (32%) |
| Decreased work | 691 (47%) | 186 (37%) | 124 (49%) |
| Change in patient contact due to the SARS-CoV-2 pandemic, n (%) | | | |
| Increased patient contact | 114 (8) | 35 (7) | 22 (9) |
| Similar patient contact | 530 (36) | 212 (42) | 90 (35) |
| Decreased patient contact | 743 (50) | 214 (43) | 119 (47) |
| No patient contact | 59 (4) | 17 (3) | 12 (5) |
| Level of COVID-19 risk of working environment reported by HCPs, n (%) | | | |
| High risk of COVID-19 infection | 1144 (77%) | 337 (68%) | 221 (87%) |
| Moderate risk of COVID-19 infection | 128 (9%) | 32 (6%) | 14 (6%) |
| Low risk of COVID-19 infection | 208 (14%) | 130 (26%) | 19 (7%) |
| Interaction with patients with suspected/confirmed SARS-CoV-2, n (%) | 601 (41%) | 121 (24%) | 145 (57%) |
| Is PPE used by the respondents? n (%) | | | |
| Not yet | 171 (12) | 75 (15) | 33 (13) |
| Yes—for aerosol-generating procedures only | 89 (6) | 28 (6) | 16 (6) |
| Yes—for all patients | 934 (63) | 334 (67) | 124 (49) |
| Yes—for suspected/positive patients only | 286 (19) | 62 (12) | 81 (32) |
| What PPE is used by the respondents? n (%) | | | |
| None | 47 (3) | 21 (4) | 8 (3) |
| Gloves only | 20 (1) | 25 (5) | 4 (2) |
| Gloves+surgical mask | 315 (21) | 105 (21) | 77 (30) |
| Gloves+surgical mask+visor | 111 (8) | 35 (7) | 6 (2) |
| Gloves+FFP3 mask | 70 (5) | 13 (3) | 12 (5) |
| Gloves+FFP3 mask+visor | 41 (3) | 8 (2) | 2 (1) |
| Gloves+FFP3 mask+visor+gown+surgical hat | 59 (4) | 24 (5) | 7 (3) |
| Patient wearing mask | 1274 (86) | 333 (67) | 234 (92) |
| Concerns regarding a lack of PPE, n (%) | | | |
| No | 398 (27) | 140 (28) | 41 (16) |
| Gloves | 176 (12 | 63 (13) | 39 (15) |
| Gown | 403 (27) | 133 (27) | 93 (37) |
| N95/FFP3 mask | 860 (58) | 239 (48) | 180 (71) |
| Surgical hat | 142 (10) | 61 (12) | 30 (12) |
| Surgical mask | 516 (35) | 189 (38) | 115 (45) |
| Visor | 379 (26) | 122 (24) | 81 (32) |
| Surgical mask (patient) | 395 (27) | 133 (27) | 81 (32) |
| Type of PPE that has been reused or recycled, n (%) | | | |
| No | 594 (40) | 205 (41) | 66 (26) |
| Gloves | 22 (1) | 9 (2) | 3 (1) |
| Gown | 166 (11) | 92 (18) | 29 (11) |
| N95/FFP3 mask | 596 (40) | 152 (30) | 138 (54) |
| Surgical hat | 59 (4) | 22 (4) | 11 (4) |

Continued

**Table 3** Continued

| | Consultants/attending | Sonographer | Trainees/residents |
|---|---|---|---|
| Surgical mask | 357 (24) | 116 (23) | 92 (36) |
| Visor | 401 (27) | 138 (28) | 82 (32) |
| Prevalence of suspected/confirmed SARS-CoV-2-based on symptoms±PCR testing, n (%) | 187 (13) | 71 (14) | 32 (13) |
| Had SARS-CoV-2 antibody test, n (%) | 259 (18) | 55 (11) | 54 (21) |
| Had positive SARS-CoV-2 antibody, n (%) | 28 (11) | 5 (9) | 4 (7) |
| Mean days of self-isolation (range) | 9.6 (0–30) | 9.7 (0–30) | 8.5 (0–30) |
| Mean days of household self-isolating (range) | 13 (1–30) | 14.3 (2–30) | 8.8 (2–14) |
| Respondents satisfied with local unit SARS-CoV-2 response, n (%) | 1043 (70) | 340 (68) | 148 (58) |
| Mean % level of respondent satisfaction with government SARS-CoV-2 response (95% CI) | 47 (46 to 49) | 57 (54 to 59) | 44 (40 to 47) |
| Anxiety, mean (95% CI)* | 6.7 (6.4 to 7.0) | 7.7 (7.1 to 8.3) | 6.9 (6.3 to 7.6) |
| 0–7 (none), n (%) | 433 (61) | 112 (52) | 75 (58) |
| 8–10 (mild), n (%) | 151 (21) | 48 (22) | 30 (23) |
| 11–21 (moderate to severe), n (%) | 131 (18) | 54 (25) | 24 (19) |
| Depression, mean (95% CI)* | 5.3 (5.0 to 5.5) | 5.5 (5.0 to 6.1) | 5.3 (4.7 to 5.9) |
| 0–7 (none), n (%) | 523 (73) | 148 (69) | 89 (69) |
| 8–10 (mild), n (%) | 126 (18) | 50 (23) | 29 (22) |
| 11–21 (moderate to severe), n (%) | 66 (9) | 16 (7) | 11 (9) |

*Numbers of Hospital and Anxiety Depression Scale (HADS) respondents lower than the main survey (n, Consultants/attending=715; Sonographer=214; Trainees/residents=129).
HCP, healthcare professional; PCR, Polymerase Chain Reaction; PPE, personal protective equipment.

Trainees reported lower satisfaction/support from their local unit (58.3%, compared with 70.5% and 68.1% in consultants and sonographers, respectively) and government (44%, compared with 47% and 57% in consultants and sonographers, respectively).

### Univariate analysis by PPE (N=2233)
Table 4 focuses on common combinations of PPE used when performing an ultrasound scan.

### COVID-19 prevalence
Of the total respondents, 2.2% reported using gloves in isolation. Twenty-nine per cent of this group reported suspected or confirmed COVID-19.

The most used combination of PPE when performing an ultrasound scan was gloves and a surgical mask (used by 22.3% of respondents). Of participants using this combination of PPE, 45.4% (226/497) reported interacting with patients with confirmed SARS-CoV-2, with 16.9% of this group reporting suspected or confirmed COVID-19.

The use of gloves with an FFP3/N95 mask instead of a surgical mask when performing ultrasound was reported by 4.3% (without visor) and 2.3% (with visor) of HCPs. Of those using gloves and FFP3/N95 mask, 10.5% reported suspected or confirmed COVID-19, compared with 3.9% of those who also used a visor.

Four per cent of all HCPs reported using 'full' PPE, 81.1% of whom worked in a COVID-19 high-risk setting. Full PPE advised for use by clinicians when interacting with a suspected or confirmed SARS-CoV-2 patient includes gloves, gown, N95/FFP3 mask, a surgical hat and visor. Of those using full PPE, 37.8% interacted at least once with patients with suspected or confirmed SARS-CoV-2. Of this group, 6.7% reported suspected or confirmed SARS-CoV-2.

Of the HCPs, 82.4% reported patients wearing a mask during clinical interactions. This was within a high-risk area of work in 76.5% cases.

### Anxiety, depression and support
HCPs who reported PPE shortages had higher rates of anxiety (45.4%) and depression (31.5%) compared with those without shortages (29.6% and 18.5%, respectively) (online supplement 11: Comparison of Hospital Anxiety and Depression scale (HADS) scores in those with (n = 788) and without personal protective equipment (PPE) shortages (n = 270) (Total n = 1058)).

Lower satisfaction/support from their local unit ranged between 66% and 75% with a mean level of government satisfaction ranging between 40% and 68% between the PPE subgroups.

### DISCUSSION
### Summary
We found that trainees and sonographers are generally more exposed to SARS-CoV-2, with sonographers seeing more patients who did not wear a mask and trainees working the most in higher risk areas. Trainees are thus

**Table 4**  Breakdown of findings by combinations of PPE (N=2233)*

| | None | Gloves only | Gloves+surgical mask | Gloves +surgical mask+visor | Gloves +FFP3 mask | Gloves +FFP3 mask+visor | Gloves +FFP3 mask +visor+gown +surgical hat |
|---|---|---|---|---|---|---|---|
| Respondents (n, %) | 76 (3) | 49 (2) | 497 (22) | 152 (7) | 95 (4) | 51 (2) | 90 (4) |
| Level of COVID-19 risk of working environment reported, n (%) | | | | | | | |
| High risk of COVID-19 infection | 50 (66) | 34 (69) | 394 (79) | 104 (68) | 60 (63) | 37 (73) | 73 (81) |
| Moderate risk of COVID-19 infection | 9 (12) | 3 (6) | 38 (8) | 16 (11) | 7 (7) | 6 (12) | 6 (7) |
| Low risk of COVID-19 infection | 17 (22) | 12 (24) | 65 (13) | 32 (21) | 28 (29) | 8 (16) | 11 (12) |
| Interaction with patients with suspected/confirmed SARS-CoV-2, n (%) | 22 (29) | 16 (33) | 226 (45) | 47 (31) | 31 (33) | 15 (29) | 34 (38) |
| Prevalence of suspected/confirmed SARS-CoV-2 based on symptoms±PCR testing, n (%) | 10 (13) | 14 (29) | 84 (17) | 17 (11) | 10 (11) | 2 (4) | 6 (7) |
| Respondents satisfied with local unit SARS-CoV-2 response, n (%) | 57 (75) | 33 (67) | 326 (66) | 112 (74) | 65 (68) | 38 (75) | 60 (67) |
| Mean % level of respondent satisfaction with government SARS-CoV-2 response (95% CI) | 63 (56 to 69) | 68 (60 to 76) | 50 (47 to 52) | 48 (43 to 53) | 42 (36 to 48) | 40 (32 to 47) | 47 (41 to 53) |

*All possible combinations are not included in this table; therefore, the denominator of the table is not 2233 (n=1010). Main relevant PPE combinations are described.
PCR, Polymerase Chain Reaction; PPE, personal protective equipment.

more likely to interact with patients with suspected or confirmed SARS-CoV-2 and have greater PPE concerns with poorer satisfaction and support. The most common PPE combination in use by HCPs were gloves and a surgical mask, with 18% of patients reporting not wearing a mask. Prevalence of SARS-CoV-2 was lower among HCPs when visors, filtering face masks or full PPE was used. Up to one in five respondents met the criteria for either moderate to severe anxiety or depression. Our findings suggest that suspected or confirmed SARS-CoV-2 prevalence among HCPs working with ultrasound in obstetrics and gynaecology was at least 13% at the time of the survey, with a higher prevalence in the UK.

### Strengths

The main strengths of our study are the relatively large sample size, the assessment of HCP responses internationally and the ability to compare responses between different HCPs. We translated the survey into four languages to include as many potential participants as possible. Furthermore, the response rate of ISUOG members was greater than non-ISUOG members, as expected, given the target population. We compared responses between ISUOG and non-ISUOG members to ensure the data could be reliably grouped for the analysis.

### Weaknesses

The key weakness of the study was the relatively low response rate and the method of sample selection that introduces possible selection bias when interpreting the results. As this is a descriptive study discussing early experience and well-being of HCPs working with ultrasound in obstetrics and gynaecology during the evolving pandemic, it is important not to draw conclusions about causation, one reason why univariate analyses were performed. For example, we cannot link use of PPE to levels of anxiety and depression in this study. We combined suspected and confirmed SARS-CoV-2, at the time of writing, as the majority of HCP did not have the same access to a swab or antibody test that is now available.

### COVID-19 prevalence

Although international prevalence varies, at the time of writing, confirmed global SARS-CoV-2 cases had surpassed 90 million. The suspected or confirmed SARS-CoV-2 prevalence of 13% in this international study is consistent with other published reports on SARS-CoV-2 and HCP, where prevalence ranges from 1.6% to 22%.[16–22] These studies have now been conducted across multiple specialities in many countries, with prevalence determined by PCR as well as antibody testing. One UK study within the field of obstetrics and gynaecology identified an HCP SARS-CoV-2 prevalence of 22%.[18] Given that the advice for use of PPE is consistent within obstetrics and gynaecology,[6] the higher prevalence may relate to respondents not using PPE or not using it correctly. The prevalence suggested by our dataset may be associated with PPE supply concerns, the need to recycle or reuse PPE, a lack

of guidance regarding when to use various levels of PPE, or due to many respondents experiencing high-risk interactions with patients, for example, scanning in proximity for prolonged periods of time.

### PPE use and provision

Countries with greatest concern for PPE from our survey included Spain, Italy, the Philippines and the USA, with similarly high rates of PPE recycling. In the UK and India, over half of respondents expressed concern. The SARS-CoV-2 pandemic led to imposed export restrictions as domestic needs increased, demand shock, rush of PPE acquisition and supply chain failures with variation of national healthcare advice.[23–26] As an example, China was responsible for mass production of clinical gowns, manufacturing 50% of the world's demand prepandemic.[23] This all contributed to the soaring global costs of PPE and the financial consequences of obtaining the necessary materials to safely support each healthcare system. In the USA, gowns were 7.5 times more expensive in 2020 compared with 2019.[24] Face masks were nine times more costly; N95 respirators were priced eight times the prepandemic rate; and gloves cost 2.5 times more.[24] PPE concern and recycling may have been a consequence of the logistical and financial complications seen as part of the SARS-CoV-2 response. However, as each healthcare system had varying prepandemic funding concerns and constraints, as well as varying HCP advice, these may have also compounded PPE supply and distribution further at a time of extreme need.

More trainee doctors reported working in higher-risk areas, as well as being more likely to interact with patients with suspected or confirmed SARS-CoV-2 than consultants or sonographers. They also had higher levels of concern regarding PPE, reusing and recycling more PPE, with a poorer perception of unit and governmental satisfaction and support. It is well documented that HCP burnout relates to anxiety and depression and that burnout is particularly prevalent among trainees.[27] Fears of infection and death during the pandemic may exacerbate this problem[28] and may also relate to inadequate support or protection.

Sonographers also reported seeing more patients who did not wear a mask. The survey shows a worrying pattern where HCPs with the least agency (trainees and sonographers) were not only more likely to see more high-risk patients but were less likely to be protected. It is perhaps not surprising they expressed greater concerns than other colleagues. PPE issues may relate to systematic financial healthcare consequences previously described but, in this instance, may also have local aetiology. Those with least agency also tend to have the least power to instigate change in their department. Concerns may never reach managers and administrators who also may not be clinically trained or orientated. Communication failure may have led to the PPE concerns expressed by HCPs.

Consensus statements and national and international guidance have been published on advised levels of PPE in

relation to SARS-CoV-2, with increasing requirements based on clinical risk. At the top end of the scale, disposable gloves, disposable fluid-repellent coverall/gown, filtering face piece respirator and eye/face protection should be used when performing aerosol-generating procedures on possible or confirmed cases of SARS-CoV-2.[6 29 30] Of the study participants, 22.3% reported using gloves and a surgical mask, the combination recommended by the ISUOG consensus statement for HCPs performing ultrasound scans.[6] Others reported using various combinations of PPE, some of which is recommended for high-risk, aerosol-generating patient–HCP interactions. However, a proportion also reported not using PPE. This variation in PPE may be due to local policy, dependent on wards or clinical areas where HCP duties extend beyond ultrasound, or a result of the PPE supply issues described previously. The use of full PPE or the addition of eye/face protection and/or filtering face piece respirators to gloves is associated with lower reports of suspected or confirmed SARS-CoV-2 when compared with gloves and surgical mask alone. This corroborates literature findings.[19 22]

## Anxiety and depression

In our study, just under half of the survey respondents completed the HADS questionnaire. This may have been due to a language barrier. However, there is also stigma related to HCPs seeking help and support, even though it is known HCPs experience high levels of psychological morbidity.[31–34] This may have discouraged HCPs to provide information relating to anxiety and depression.

From those who completed the HADS, almost half the respondents reported some level of anxiety and depression, with 19.8% and 8.8% experiencing moderate-to-severe anxiety and depression, respectively. Greater proportions were among female participants, with more depression reported by those working in lower-risk areas. Again, this may relate to the stigma for HCPs reporting psychological morbidity and seeking support, but also may be explained by a completion bias, that is, those with psychological morbidity may be more inclined to complete the HADS to increase awareness of these issues.[31–34]

Given the design of this study, we must be careful not to relate this to COVID-19 and be aware that there may be selection bias. There are aspects of the obstetrics and gynaecology environment which may encourage higher pre-existing levels of anxiety and depression among HCPs prior to the SARS-CoV-2 pandemic. The investigating traumatic work-related events in obstetrics and gynaecology (INDIGO) study reports two-thirds of obstetrics and gynaecology trainees and consultants in the UK have been personally traumatised by work-related events unrelated to SARS-CoV-2, with 31% of them affected by post-traumatic stress disorder symptoms.[35] Of the total paediatricians, 14.1% and 7.3% reported mild to severe anxiety and depression using HADS in a survey published prior to the pandemic,[36] a specialty that reflects similar intensity.

Our findings strongly support the need for reliable infrastructure that provide HCP counselling and psychological support without stigma in every medical and surgical specialty. In the UK, services such as NHS Practitioner Health and the British Medical Association offer confidential support and protect HCPs at the national level, while locally, seniors provide regular debriefing sessions. However, provision of local and global support is not consistent, and thus movement beyond a culture of HCP stigma is essential.[31]

## CONCLUSION

This study provides insight into the experience and well-being of clinicians working in the field of ultrasound in obstetrics and gynaecology during the early phase of the SARS-CoV-2 pandemic. Although further work is required to unpick the associations of the pandemic, PPE availability, SARS-CoV-2 prevalence, and anxiety and depression, we hope our report highlights the importance of provision of PPE and the need for national and international consistencies in advice regarding PPE requirements, regulations and use. HCPs all take risk daily, particularly trainees and sonographers who are generally more exposed to SARS-CoV-2 and less unable to communicate their concerns and needs. As many patients may harbour infection with minimal or without symptoms, there is a real need to ensure HCPs are consistently and adequately supported.

**Author affiliations**
[1]Early Pregnancy and Acute Gynaecology Unit, Department of Obstetrics and Gynaecology, Queen Charlotte's and Chelsea Hospital, Imperial College London, London, UK
[2]Department of Development and Regeneration, KU Leuven, Leuven, Belgium
[3]Department of Oncology, Laboratory of Tumour Immunology and Immunotherapy, KU Leuven, Leuven, Belgium
[4]Facultad de Medicina, Clínica Alemana, Universidad del Desarrollo, Santiago de Chile, Chile
[5]Facultad de Medicina, Clínica Alemana, Hospital Clínico de la Universidad de Chile José Joaquín Aguirre, Santiago, Chile
[6]Département d'échographie en Gynécologie et Obstétrique, Centre d'Échographie de l'Odéon, Paris, France
[7]Dipartimento Scienze della Salute della Donna, del Bambino e di Sanità Pubblica, Fondazione Policlinico Universitario Agostino Gemelli IRCCS, Roma, Italy
[8]Centre for Fetal Care, Department of Obstetrics and Gynaecology, Queen Charlotte's and Chelsea Hospital, Imperial College London, London, UK
[9]Department of Obstetrics and Gynaecology, KU Leuven University Hospitals Leuven, Leuven, Belgium

**Contributors** TB, CK, HS and DT participated in the conception and design of the study. TB, CK and HS created the English version of the survey. JP, UM and CL translated survey to Spanish, French and Italian. JC performed the statistical analysis. TB, CK, HS, JC and DT interpreted the results. TB, CK and HS wrote the initial version of the manuscript. All authors critically revised the manuscript and approved the final version. TB is the author acting as guarantor.

**Funding** TB is supported by the National Institute for Health Research Imperial Biomedical Research Centre (grant number IS-BRC-1215–20013). CK is supported by Imperial Health Charity (grant number RFPrD1920/116). DT is fundamental clinical researcher of Research Foundation – Flanders (FWO). JC and DT are supported by FWO (grant number G0B4716N) and Internal Funds KU Leuven (grant number C24/15/037).

**Competing interests** None declared.

**Patient consent for publication** Not applicable.

**Ethics approval** This study involves human participants. Ethical approval for the distribution of this online survey was obtained from the ethics committee of KU Leuven, Belgium (S-64016). Participants gave informed consent to participate in the study before taking part. A copy of the protocol can be viewed in online supplemental material 2.

**Provenance and peer review** Not commissioned; externally peer reviewed.

**Data availability statement** All data relevant to the study are included in the article or uploaded as supplemental information. No additional data available.

**ORCID iDs**
Tom Bourne http://orcid.org/0000-0003-1421-6059
Christopher Kyriacou http://orcid.org/0000-0001-9001-5545
Harsha Shah http://orcid.org/0000-0003-3866-7946
Chiara Landolfo http://orcid.org/0000-0001-9808-7957
Christoph Lees http://orcid.org/0000-0002-2104-5561
Dirk Timmerman http://orcid.org/0000-0002-3707-6645

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
