## [Reviewer comments · BMJ Open]

ARTICLE DETAILS

TITLE (PROVISIONAL)	The experiences and wellbeing of healthcare professionals working in the field of ultrasound in Obstetrics and Gynecology as the SARS-CoV-2 pandemic was evolving: a cross-sectional survey study
AUTHORS	Bourne, Tom; Kyriacou, Christopher; Shah, Harsha; Ceusters, Jolien; Preisler, Jessica; Metzger, Ulrike; Landolfo, Chiara; Lees, Christoph; Timmerman, Dirk

VERSION 1 – REVIEW

REVIEWER	Chervenak, F Department of Obstetrics and Gynecology, Zucker School of Medicine of Hofstra/Northwell, Lenox Hill Hospital
REVIEW RETURNED	04-May-2021

GENERAL COMMENTS	This paper makes a valuable contribution to the literature by providing an international survey from 115 countries in four languages assessing psychological and physical well being and anxiety and depression. multiple important findings reported Paper is very timely, well-written, and easy to follow
--

REVIEWER	Scanlan, Gillian University of Dundee, Centre for Medical Education
REVIEW RETURNED	25-May-2021

GENERAL COMMENTS	T thank you for your submission, on an important and critical area to explore in medical education. I look forward to your next iteration and I hope the comments below are helpful in structuring the next version. Abstract- background or introduction to the study would have been beneficial. Outlining the problem and the context in which the study was taking place. Add cross sectional survey design. The primary outcome needs expanded, for example in the main text you have this 'The primary outcome was the overall prevalence of SARS-CoV-2, depression and anxiety amongst HCPs in relation to country and PPE availability'. The specific type of analysis that was undertaken needs to be included e.g. descriptive stats and univariate analysis was used to address the studies objectives. How many mailing list participants were contacted for the study? Methods: Refer the reader to the physical aspects section of the questionnaire, include as supplementary file and refer reader to this
--

	information. How many participants chose not to complete the HADs? Why do you think this was the case? Missing data was not included in the analysis? What type of missing data was this? This could be briefly outlined in text. Analysis/results In which platform was the data analysed using? You indicate that age was recorded by you do not report this in the demographics survey information found in table 1. In the written text found on page 10 ensure to include the figure for male and females within the survey data. Did you collect any other socio-demographic information? If so, why is this not reported in text or in the table? For example, age mean is included but did you not have set parameters for each age group this should be reported and how many were in each group. It would have been useful to conduct multivariate analysis on this data, you split participants into specific groupings (consultant/attending, sonographers and trainees/residents) a further exploration of these groups seems like a missed opportunity. Was there certain groups who experienced higher levels of anxiety and depression that others was this related to stage of career/gender/PPE/risk factors etc? Those with a high risk of COVID-19 infection where they more likely to experience anxiety and depression that others? In the discussion you indicate 'this is a descriptive study discussing early experience and wellbeing of HCP working with ultrasound in obstetrics and gynaecology during the evolving pandemic, it is important not to draw conclusions about causation. For example, we cannot link use of PPE to levels of anxiety and depression in this study'. However, I am unclear why this analysis was not undertaken as it is likely an important area of exploration? I think this is area that could have strengthened your research and provided some potential future avenues of empirical data collection. Discussion When considering your findings- I would encourage you to explore the PPE issues in more detail. For example, some countries may have had PPE shortages due to the financial issues within their healthcare sector? There is a lot of time spent on reiterating the PPE findings from the main study, I would encourage your research team to consider thinking about what these main findings mean? What recommendations would you make based on these findings? Does messaging around PPE need to be consistent? Each country took different decisions based on the pandemic, do you think this could have impacted the use of PPE in some contexts etc? The HADS scale- some participants chose not to complete this stage of the survey. This could have been explored briefly in the discussion. Why did you think some opted out? Do you think it's related to stigma around mental health for healthcare professionals? There has been a considerable amount of work focussing upon these issues. In the 'anxiety and depression' section in the discussion it is mentioned that almost half of participants reported some level of anxiety and depression in your study. Based on this finding, what support mechanisms do you think need to be in place for HCPs working in this specialty? In discussion/conclusion section discuss how your findings could be used in the field? What recommendations would you make to protect the physical and mental wellbeing of hcp's working in obs/gyn? Do
--	---

	your messages convey to other contexts and or disciplines? Is there any unanswered questions that your study has risen? How you considered areas for future research? Is there an area for qualitative exploration to unpack the issues outlined in your findings? Could you explore the differences amongst groups in more detail?
--	--

VERSION 1 – AUTHOR RESPONSE

Reviewer: 1

Dr. F Chervenak, Department of Obstetrics and Gynecology, Zucker School of Medicine of Hofstra/Northwell, Lenox Hill Hospital

Comments to the Author:

This paper makes a valuable contribution to the literature by providing an international survey from 115 countries in four languages assessing psychological and physical well being and anxiety and depression.

multiple important findings reported

Paper is very timely, well-written, and easy to follow

Many thanks for your feedback.

Reviewer: 2

Dr. Gillian Scanlan, University of Dundee

Comments to the Author:

Thank you for your submission, on an important and critical area to explore in medical education. I look forward to your next iteration and I hope the comments below are helpful in structuring the next version.

Many thanks for your comments and feedback.

Abstract-

background or introduction to the study would have been beneficial. Outlining the problem and the context in which the study was taking place.

Many thanks for your comment. We have added more information to the objectives section of the abstract to outline the problem and context further.

Add cross sectional survey design.

Many thanks. We have added this to the design section of the abstract.

The primary outcome needs expanded, for example in the main text you have this ‘The primary outcome was the overall prevalence of SARS-CoV-2, depression and anxiety amongst HCPs in relation to country and PPE availability’.

Many thanks. We have elaborated further in this section of the abstract.

The specific type of analysis that was undertaken needs to be included e.g. descriptive stats and univariate analysis was used to address the studies objectives.

Many thanks. This has been added as a new subsection within the abstract.

How many mailing list participants were contacted for the study?

Many thanks for your comment. The survey was sent to the mailing list of the International Society of Ultrasound in Obstetrics and Gynecology (ISUOG) which is comprised of 35,509 health care professionals (HCPs) in 124 countries This has been added to the abstract.

Methods:

Refer the reader to the physical aspects section of the questionnaire, include as supplementary file and refer reader to this information.

Many thanks for your comment. We had already included the questionnaire in its entirety in the supplementary material (Supplementary Material 1). However, we have also made it clearer in the text where to obtain this information and which questions reflect physical aspects (questions 2-40), and which questions relate to the HADS (questions 41-56).

How many participants chose not to complete the HADs? Why do you think this was the case?

Many thanks for your queries. We have attended to the answer of the first question in the results section of the manuscript. 1058/2237 (47%) completed the HADS. The remainder (1179/2237, 53%) chose not to complete the HADS. We do not know for sure why they chose not to complete it, as we were not collecting data to explain this. It may have been due to a language barrier, or the fact that HCPs did not feel comfortable divulging this information. We have contemplated this point in the discussion section of the updated manuscript.

Missing data was not included in the analysis? What type of missing data was this? This could be briefly outlined in text.

Many thanks for your queries. Some participants chose to complete the questionnaire but to skip some of the questions. We therefore excluded any questions that were unanswered from the analyses. This has been outlined in the updated manuscript.

Analysis/results

In which platform was the data analysed using?

Many thanks for your question. For the data analysis, R, version 3.5.1, was used. This information has been added to the methods section of the manuscript.

You indicate that age was recorded by you do not report this in the demographics survey information found in table 1.

Many thanks for your comment. Age is recorded on the thirteenth row of Table 1. Mean is 47.2 years, with a range of 18-82 years. However, we have now elaborated on age in the updated Table 1, where we have derived age within seven categorical age groupings. We have added the equivocal information to Supplementary Material 6, as well as further describing age-related findings in the results section.

In the written text found on page 10 ensure to include the figure for male and females within the survey data.

Many thanks for your comment. We have added to the information presented in the paragraph on page 10, now stating that 1474 (66%) of respondents were female and 755 (34%) were male, with 4 intersex or undisclosed.

Did you collect any other socio-demographic information? If so, why is this not reported in text or in the table? For example, age mean is included but did you not have set parameters for each age group this should be reported and how many were in each group.

Many thanks for your queries. We collected data regarding ISUOG membership which we have previously described above, in the results, in Table 1 and in more detail within Supplementary Material 6.

We also collected information on country of practice. As the responses were from 115 countries, with many only having a few responses, we discussed countries in more detail in Table 2 and in the univariate analysis by country section of the results. We provided a breakdown of the six countries that had the most respondents in order to perform a meaningful analysis and report appropriately on the data available.

We collected information on type of HCP and this data is discussed in Table 3 and in the univariate analysis by HCP section of the results.

We used the age data collected to calculate mean and range. We have since subdivided age groups and now present age data in subgroups: <20, 21-30, 31-40, 41-50, 51-60, 61-70, >70 years. We have also elaborated on these findings in the results section of the manuscript.

Our data regarding gender has also been described in more detail in the first paragraph of the results. Data relating to comorbidities are described in the summary section of the results and in Supplementary Material 7.

We did not collect any other demographical information.

It would have been useful to conduct multivariate analysis on this data, you split participants into specific groupings (consultant/attending, sonographers and trainees/residents) a further exploration of these groups seems like a missed opportunity.

Many thanks for your comment. There are two reasons why a multivariate analysis was not performed.

First, given the quantity of data from the number of questions proposed to HCPs, overlap of information made it difficult to extrapolate in a productive, useful and interpretable manner. We therefore defined three important topics of interest that allowed maximal independent data interpretation which were deemed crucial in evaluating the impact of the pandemic: country, type of HCP and PPE.

After splitting consultants/attending and trainees/residents by age, 33 HCP groupings were now in existence (Supplementary Material 3). Individual assessment at this point would have unfortunately been limited by the numbers. We therefore used three specific HCP groupings to allow data comparison of HCP categories that were likely to have objective differences in proximity and time spent with patients, important when comparing COVID-19 prevalence, PPE use and provision and psychological morbidity.

The second reason why a multivariate analysis was not performed relates to the anxiety and depression findings. A multivariate analysis would lead to causal conclusions. However, the design of the study doesn't allow a causal interpretation:

- We have no idea about the anxiety and depression levels of the participants before the SARS-CoV-2 pandemic. Therefore, it is not possible to say that the observed anxiety and depression is caused by the SARS-CoV-2 pandemic.
- Countries were in a different phase of the pandemic at the time of the survey.

We have elaborated on the above in the methods and discussion section of the updated manuscript.

Were there certain groups who experienced higher levels of anxiety and depression than others; were these related to stage of career/gender/PPE/risk factors etc?

Many thanks for your question. We hypothesized that stage of career would influence levels of anxiety and depression. However, rates were similar across HCP groups (defined as discussed in our previous comment), with rates of mild to severe anxiety 39% amongst consultants/attending, 47% in sonographers and 42% in trainees/residents (Table 3). Mild to severe depression rates were 27% amongst consultants/attending, 30% in sonographers and 31% amongst trainees/residents (Table 3). The differences between HCP groups were less pronounced when evaluating moderate to severe anxiety and depression. Given the stressful environment associated with the specialty, may be secondary to the profession and unrelated to the pandemic. We unfortunately do not know for certain and have thus presented this data with caution. Nevertheless, this has been described in the results and further elaborated in the discussion.

Regarding gender, we have added to our results with the below table, now Supplementary Material 9. A greater proportion of female participants scored for moderate to severe anxiety and depression compared to male participants. However, we do not know if this is due to the pandemic, or pre-existing mental health conditions. We have elaborated further in the discussion of the updated manuscript.

Supplement 9: HADS breakdown by gender (n = 1058).

	Female	Male	Prefer not to say
Number of respondents (n, %)	754 (71%)	303 (29%)	1 (0%)
Anxiety (mean, 95% CI)*	6.9 (6.6-7.2)	5.7 (5.3-6.2)	-
0-7 (None) (n, %)	450 (60%)	202 (67%)	1 (100%)
8-10 (Mild) (n, %)	148 (20%)	60 (20%)	0 (0%)
11-21 (Moderate to severe) (n, %)	156 (21%)	41 (14%)	0 (0%)
Depression (mean, 95% CI)*	5.4 (5.2-5.7)	5.0 (4.6-5.4)	-
0-7 (none) (n, %)	529 (70%)	230 (76%)	1 (100%)
8-10 (mild) (n, %)	146 (19%)	59 (19%)	0 (0%)
11-21 (moderate to severe) (n, %)	79 (10%)	14 (5%)	0 (0%)

Supplementary Material 11 attends to the issue of PPE. HCPs that reported PPE shortages had higher rates of anxiety and depression. 45% experienced mild to severe anxiety with PPE shortages,

compared with 30% without. 31% experienced mild to severe depression with PPE shortages, compared with 18% without. Similar proportional differences were seen when evaluating moderate to severe anxiety and depression only. This is described in the results and the discussion.

We also evaluated psychological morbidity based on country which are described in the results. Country comparisons were performed to describe psychological data with COVID-19 prevalence and PPE use and provision. For mild to severe anxiety and depression, rates were: 28 and 29% in Italy, 39 and 20% in Spain, 49 and 28% in India, 50 and 25% in the Philippines, 42 and 29% in the UK, 41 and 22% in the US, and 43 and 30% elsewhere. When focussing on just moderate to severe anxiety and depression, rates were: 16 and 10% in Italy, 14 and 7% in Spain, 19 and 5% in India, 22 and 3% in the Philippines, 23 and 10% in the UK, 27 and 7% in the US, and 20 and 10% elsewhere (Table 2).

Unfortunately, in each case, we cannot know whether the rates relate to the pandemic and PPE or whether confounders such as the stress of working in Obstetrics and Gynecology influence our findings based on career, gender, PPE, or country. We have elaborated on this further in our discussion.

Those with a high risk of COVID-19 infection: were they more likely to experience anxiety and depression than others?

Many thanks for your question. The total numbers are as follows: 1702/2233 (76%) working in a high-risk environment, 174/2233 (8%) working in a moderate risk environment and 357/2233 (16%) working in a low-risk environment (Table 3). When focussing on those that completed the HADS (n = 1058): 23/1058 (2%) work in low-risk settings, 28/1058 (3%) in moderate-risk settings and 1007/1058 (95%) in high-risk settings (Supplementary Material 10).

Although the proportion of HCP working in low, moderate and high-risk areas are generally similar when total responses are considered (n = 2233), those working in high-risk areas appear more inclined to complete the HADS. However, the rates of mild to severe anxiety are overall similar, with more depression reported by those working in lower risk areas.

This may be explained by a completion bias i.e., those with psychological morbidity may be more inclined to complete the HADS. However, we cannot know for certain with the data collected. We have elaborated on these findings in our results and discussion.

Supplement 10: HADS breakdown by risk of working area (n = 1058).

	Low risk	Moderate risk	High risk
Number of respondents (n, %)	23 (2%)	28 (3%)	1007 (95%)
Anxiety (mean, 95% CI)*	8.3 (6.3-10.3)	5.1 (3.6-6.6)	6.6 (6.3-6.8)
0-7 (None) (n, %)	13 (57%)	21 (75%)	619 (61%)
8-10 (Mild) (n, %)	3 (13%)	3 (11%)	202 (20%)
11-21 (Moderate to severe) (n, %)	7 (30%)	4 (14%)	186 (18%)
Depression (mean, 95% CI)*	8.2 (6.0-10.3)	4.3 (2.7-5.8)	5.3 (5.0-5.5)
0-7 (none) (n, %)	12 (52%)	19 (68%)	729 (72%)
8-10 (mild) (n, %)	4 (17%)	6 (21%)	195 (19%)
11-21 (moderate to severe) (n, %)	7 (30%)	3 (11%)	83 (8%)

In the discussion you indicate ‘this is a descriptive study discussing early experience and wellbeing of HCP working with ultrasound in obstetrics and gynaecology during the evolving pandemic, it is important not to draw conclusions about causation. For example, we cannot link use of PPE to levels of anxiety and depression in this study’. However, I am unclear why this analysis was not undertaken as it is likely an important area of exploration? I think this is area that could have strengthened your research and provided some potential future avenues of empirical data collection.

Many thanks for your comment. The over-riding issue when assessing psychological morbidity, especially anxiety and depression, is that although some participants may directly feel this way as a result of PPE, are they also feeling this due to lack of SARS-CoV-2 testing, lack of senior support, burnout, stress related to working in Obstetrics and Gynecology, or is there also pre-existing anxiety

and depression? There are so many reasons, even before the pandemic, why respondents would score highly for anxiety and depression. As a result, although the data is interesting, we cannot imply direct causation. We have therefore been cautious when drawing conclusions and described the evidence in an unbiased manner but have explored this further in the updated discussion section of the manuscript.

Discussion

When considering your findings- I would encourage you to explore the PPE issues in more detail. For example, some countries may have had PPE shortages due to the financial issues within their healthcare sector? There is a lot of time spent on reiterating the PPE findings from the main study, I would encourage your research team to consider thinking about what these main findings mean? What recommendations would you make based on these findings? Does messaging around PPE need to be consistent? Each country took different decisions based on the pandemic, do you think this could have impacted the use of PPE in some contexts etc? Many thanks for your comments and questions. Indeed, various countries would have had PPE shortages due to financial constraints. However, as we did not collect data directly regarding financial concerns, we cannot directly correlate these findings. However, we have elaborated on the issues and discussed this further in the updated manuscript.

We have also added more detail describing what we hypothesize the findings to mean in relation to countries, HCPs and PPE and have linked this to guidelines and consensus statements.

We had not provided any recommendations based on our findings as this is a descriptive study and thus cannot draw causative conclusions. We can, however, provide loose recommendations and have elaborated on these in the concluding paragraph of the updated manuscript.

The HADS scale- some participants chose not to complete this stage of the survey. This could have been explored briefly in the discussion. Why did you think some opted out? Do you think it's related to stigma around mental health for healthcare professionals? There has been a considerable amount of work focussing upon these issues.

Many thanks for your comments and questions. We agree that there is considerable stigma that remains in healthcare regarding seeking help. However, the language barrier may have made it difficult for some respondents, but the imbalance of who responded may be related to a completion bias. Some respondents who have a mental health history may be more inclined to complete the HADS to increase awareness and exposure of such issues. We have elaborated on these points in the updated manuscript.

It the 'anxiety and depression' section in the discussion it is mentioned that almost half of participants reported some level of anxiety and depression in your study. Based on this finding, what support mechanisms do you think need to be in place for HCPs working in this specialty? In discussion/conclusion section discuss how your findings could be used in the field? What recommendations would you make to protect the physical and mental wellbeing of hcp's working in obs/gyn? Do your messages convey to other contexts and or disciplines?

Many thanks for your comments and questions. We have elaborated in the discussion regarding how our findings should be used and what support we believe is required across all medical and surgical specialties. Our findings strongly support the need for reliable infrastructure that provide HCP counselling and psychological support without stigma. We have also described current local and national UK services that protect HCPs as examples. However, provision of local and global support is not consistent and thus movement beyond a culture of HCP stigma is essential. There is a real need to drive such support and we hope this paper contributes to this cause.

Is there any unanswered questions that your study has risen? How you considered areas for future research? Is there an area for qualitative exploration to unpack the issues outlined in your findings? Could you explore the differences amongst groups in more detail?

Many thanks for your comments and questions. Future research aims to detail associations between the pandemic and PPE availability, SARS-CoV-2 prevalence and PPE availability, and SARS-CoV-2 prevalence, PPE availability and psychological morbidity. These must be evaluated independently to control for confounders and allow groups to be explored in more detail than was possible with our current study. Qualitative exploration would be best utilised in the instance of evaluating psychological morbidity as it would allow HCPs to express how they feel beyond the measurements of an objective multiple choice tool. We have summarised the above within the concluding paragraph of the updated manuscript.